# Experimental Investigation of Additive Manufacturing Using a Hot-Wire Plasma Welding Process on Titanium Parts

**DOI:** 10.3390/ma14051270

**Published:** 2021-03-07

**Authors:** Pattarawadee Poolperm, Wasawat Nakkiew, Nirut Naksuk

**Affiliations:** 1Graduate Program in Industrial Engineering, Department of Industrial Engineering, Faculty of Engineering, Chiang Mai University, Chiang Mai 50200, Thailand; pattarawadee_pool@cmu.ac.th; 2Advanced Manufacturing Technology Research Center (AMTech), Department of Industrial Engineering, Faculty of Engineering, Chiang Mai University, Chiang Mai 50200, Thailand; 3Department of Industrial Engineering, Faculty of Engineering, Chiang Mai University, Chiang Mai 50200, Thailand; 4Automation for Material Processing Research Team, Material Processing and Manufacturing Automation Research Group, National Metal and Materials Technology Center (MTEC), National Science and Technology Development Agency (NSTDA), Thailand Science Park, Pathum Thani 12120, Thailand; nirutn@mtec.or.th

**Keywords:** hot-wire, plasma welding, additive manufacturing, wire feed, titanium alloy, mechanical properties

## Abstract

In this paper, we propose hot-wire plasma welding, a combination of the plasma welding (PAW) process and the hot-wire process in the additive manufacturing (AM) process. Generally, in plasma welding for AM processes, the deposit grain size increases, and the hardness decreases as the wall height increases. The coarse microstructure, along with the large grain size, corresponds to an increase in deposit temperature, which leads to poorer mechanical properties. At the same time, the hot-wire laser process seems to contain an overly high interstitial amount of oxygen and nitrogen. With an increasing emphasis on sustainability, the hot-wire plasma welding process offers significant advantages: deeper and narrow penetration than the cold-wire plasma welding, improved design flexibility, large deposition rates, and low dilution percentages. Thus, the hot-wire plasma welding process was investigated in this work. The wire used in the welding process was a titanium American Welding Society (AMS) 4951F (Grade 2) welding wire (diameter 1.6 mm), in which the welding was recorded in real time with a charge-coupled device camera (CCD camera). We studied three parameters of the hot-wire plasma welding process: (1) the welding speed, (2) wire current, and (3) wire feeding speed. The mechanical and physical properties (porosity, Vickers hardness, microstructure, and tensile strength) were examined. It was found that the number of layers, the length and width of the molten pool, and the width of the deposited bead increased, while the height of the layer increased, and the hot-wire current played an important role in the deposition. In addition, these results were benchmarked against specimens created by a hot-wire plasma welding/wire-based additive manufacturing process with an intention to develop the hot-wire PAW process as a potential alternative in the additive manufacturing industry.

## 1. Introductions

Additive manufacturing (AM) describes technologies that are used to fabricate 3D objects by adding a cross-sectional layer upon a layer of liquid, powder, wire, or sheet material, whether the material is metal, polymer, or others. AM is currently being used to develop and customize end-use products in several industries. The relatively low cost for the low-volume development of near-net-shape components, lower net processing expenses, shorter lead times, and the construction of more complicated geometries are some advantages of using AM technologies over conventional tooling-based methods such as casting or machining.

At present, several welding techniques are applied to AM processes, such as gas tungsten arc welding (GTAW) [1,2,3,4,5], electron beam melting (EBM) [6,7], gas metal arc welding (GMAW), laser welding [8,9,10,11], and plasma welding (PAW) [12]. The principle of the hot-wire process is to heat the filler wire (or passing current), which makes the filler wire heat up by a separate power source until it is close to the melting point, and feed hot-wire to the weld pool. Prior to reaching the welding pool, the filler wire is heated resistively. This provides benefits in terms of metallurgical control, energy efficiency, and the rate of deposition. Importantly, the filler wire can be melted independently of the arc heat source that melts the base metal [13]. The filler wire is melted by current heating from the hot-wire system, independently of the deposition rate from the arc current. The hot-wire addition from [14] can achieve a low dilution ratio, a narrower HAZ width, and a higher weld metal hardness. Moreover, increasing the hot-wire current and the wire feed rate also provides for a better and desired weld pool [15]. Based on the investigations, combining hot-wire feeding with the welding process appears to have the potential to achieve high efficiency and welding faster; in other words, the combination of higher hot-wire feeding and a lower welding current resulted in a much lower heat input while maintaining the same deposited metal volume achieved using a higher welding current. Moreover, the bead geometry was described in terms of width, penetration depth, and the area of fusion. In [16], on Consumable Electrode Arcless Electric Working, it is found that the hot-wire process is a forming metal deposited by passing an arc current through it, and the filler wire heats up. The filler wire is melted and fed heat by its own hot-wire machine, so the filler wire can be melted independently of the arc heat source, which melts the base metal. The deposition of the material is carried out layer by layer and finished when the geometry is developed [17]. The titanium alloy also has a high specific strength, a high fracture toughness, an excellent corrosion resistance, and a low density, which leads to its wide applicability in the manufacturing of parts [18,19]. In recent years, there have been many studies [15,20,21] on hot-wire GTAW, hot-wire GMAW, and hot-wire laser welding; most of the works are about suitable parameters and their effect on the quality of the workpiece received, the mechanical properties, and the penetration depth. The penetration depth of the metal is in the layer-by-layer deposited weld, and the temperature of the surrounding heat zone higher. Selecting unsuitable parameters can result in welding defects. The effect of vibration and hot-wire gas tungsten arc is on the geometry. Cold-wire GTAW and the vibrating wire feeder decreased the droplet release time, while no significant difference in the weld bead geometry was observed in cold-wire or hot-wire GTAW [5]. Pai et al. [22] studied the mechanical properties (tensile, hardness, and bend tests) to compare the results between the use of hot-wire and cold-wire GTAW. The results showed that the hardness, bend, and tensile properties of hot-wire and cold-wire GTAW processes were similar, and there was no significant difference between them. The hot-wire GTAW process has the advantage of high quality with an excellent weld profile and high productivity. At the same time, Naksuk et al. [23] studied the temperature generated during welding recorded in real time with a high-speed infrared thermography camera for the hot-wire plasma arc welding process.

AM via hot-wire PAW freeform welding, which is the main focus of this study, can investigate the influence of the hot-wire PAW technique of titanium alloy on the AM process. Moreover, the high content of oxygen and nitrogen formed after part forming by AM was the main cause of cracking in the hot-wire laser welding process. The work of [15] was conducted to study the suitable parameters of hot-wire GTAW, such as the welding current, the hot-wire current, and the wire feed rate. The results obtained indicate that, for a constant welding current, increasing the hot-wire current and wire feed rate provides stable voltage–current characteristics and a higher bead weight. The ordinary parameters of the hot-wire process, such as the wire current, wire feed rate, wire size, wire feed angle, and gas shield, have an important effect on the quality of the workpiece received [20,21]. While research papers on the hot-wire process are available [24,25,26], there is limited research on hot-wire plasma welding. Therefore, it is important to study the basic parameters of hot-wire plasma welding to obtain good mechanical properties in a titanium welding workpiece. The definitions of good mechanical properties of titanium alloy grade 2 provided an excellent balance of medium strength and reasonably good ductility.

## 2. Materials and Experimental Procedure

### 2.1. Experimental Material and Equipment

Titanium alloy plates of grade 2 in the additive manufacture with the dimensions of 135 mm (width) × 350 mm (length) × 10 mm (thickness) were welded in this study. The substrate materials used a titanium alloy grade 2, the dimensions of 135 mm × 350 mm (width × length), and a thickness of 10 mm. These have a chemical composition measured using an energy-dispersive X-ray spectroscopy (EDS) machine (Shimadsu, Kyoto, Japan) with a standard chemical composition and mechanical physical [27,28,29] values as in Table 1 and Table 2.

The filler wire was a titanium AMS 4951F (Grade 2) welding wire with a diameter of 1.6 mm in the welding process, which is a filler metal process by the American Welding Society (AWS, Miami, FL, USA). The workpiece was securely clamped by jigs and fixtures to prevent it from moving or from deflection during the hot-wire plasma welding process. The plasma arc welding torch was attached to a 6-axis ABB robot of freedom, which was used to generate the movement of the welding torch relative to the reference point on the substrate. The articulated robots, feature six axes, also called six degrees of freedom. The 6-axis robots allow for greater flexibility than robots with fewer axes. It also allows freedom of movement in three-dimensional space. The robot is the handle of the torch head only. Thus, we do not have to build up the Z-axis to facilitate the movement and maintain the arc distance of the robot. The movement in the additive build direction is performed by the robot and the writing codes to control the direction of the robot. Robot studio is software developed by ABB robot that writes a working sequence for robots. The power source was used to begin the process of welding and was operated with the robot in tandem.

Figure 1 shows the setup of the hot-wire plasma welding system, which includes a robot control cabinet, a plasma arc welding torch, a hot-wire machine, a monitoring system consisting of a CCD camera, and an ABB limited robot.

Plasma welding also uses hot wires in the depositing process. The plasma welding controller from the Cebora machine brand (Bologna, Italy) was used for controlling various parameters for plasma welding. The hot-wire unit used in this research is from MAC brand, Power assist IV-642 model (Osaka, Japan) [23], with a wire feed system to the welding torch, which is attached to the robotic arm used for controlling the hot-wire plasma welding path. The plasma torch was at the front, while the hot-wire was fed behind the plasma welding torch. The experimental setup of the hot-wire unit depends on plasma welding parameters, where the arc current and welding speed of the robot are constant. We then adjusted the relationship of the wire current and the wire feed speed so that the added filler wire can move continuously in the melting pool. The hot-wire PAW process was carried out under the inter gas (argon) shield.

### 2.2. Methodology

The designs of the full factorial experiments used in this study were based on three factors, (1) the welding speed [mm/s], (2) the wire current [A], and (3) the wire feeding speed [m/min] using a significant level, α = 0.05, by strictly controlling the other factors at the same value and condition for hot-wire plasma welding.

Thus, in this experiment, there were a total of 2^3^ = 8 trials. An example of an image obtained from the results of the designed experiment is shown in Table 3 and Figure 2.

The single bead number 2.1 had a better appearance and shape than the other weld beads and met the requirements. In addition to the above criteria, we found that the filling wire was not a problem, with no interruption, no splashes, and a smooth movement. The wire tip did not stick to the ceramic tip while the wire was pulled back (ceramic rod: wire support). Thus, the welding condition of single bead number 2.1 was applied for the deposited wall of the hot-wire PAW process. The hot-wire plasma welding parameters used in this research are shown in Table 4.

Titanium is a metal that is sensitive to high-temperature oxidation. Titanium welding is required under the suitable shielding gas or inside a covered gas box. A shielded environment is required during the hot-wire plasma welding process for the wire + arc additive manufacture of titanium alloys to prevent oxidation and cracks. If air interacts with the weld, an oxide is formed with a color that depends on the thickness of the oxide layer. The acceptability level of the color depends on the application. The workpiece is found to have oxidation during the build of the first weld. Applying the shielding can increase the flexibility of the process. An example of a flow-purge welding chamber was used in the hot-wire PAW process shown in Figure 3. The positions of the weld bead height and bead width measurement are shown in Figure 4.

The welding process started from the right-hand side of the workpiece and ended at the left-hand side, as shown in Figure 5. The dimensions of the welded wall structures were 13.25 mm (width), 300 mm (length), and 63.2 mm (height).

#### 2.2.1. Porosity Analysis

The current detection of the inside pore and other defects are frequently per-formed using the X-ray method after welding. This method is an X-ray application with computed tomography (XCT), which is a high-resolution system for 2D X-ray inspection, 3D computed tomography, and the measurement of various metal materials from GE Phoenix, system: V|tome|X S (Boston, MA, USA) using a high-power nano focus X-ray tube.

#### 2.2.2. Vickers Microhardness

Vickers microhardness tests for specimens were carried out with a diamond-shaped pyramid head. The pyramid’s top angle was 136 degrees. This test can measure the hardness of a very soft metal, about 5 Kgf/mm^2^, or that of a very stiff metal, about 1500 Kgf/mm^2^, without changing the indenter. It only changes the pressure. The values between 1 and 120 Kgf depend on the hardness of the metal [30].

The wall was cut into three sections. A schematic illustration of micro-hardness measurement points on the profile of the wall, with the microhardness of the layer bands (top, middle, and bottom region) of the wall, is shown in Figure 6. Each position was measured thrice; an average of these measurements was then calculated and is graphically displayed. The cut samples of all three specimens underwent grinding and polishing and became the cross-section area of the workpiece. During grinding, the workpiece was lubricated with water.

The instrument used was a Microhardness Tester Anton-Paar, MHT-10, S/N 240826 (Anton Paar, Graz, Austria) The test conditions of the hardness test were performed under the ASTM specs standard test method E92 [31], with a US specs standard test method AMS 4951 welding wire [32]. The wall was cut into sections at three positions. In [33], square samples, such as the top, the middle, and the bottom region of the wall, were taken in the direction of the horizontal construction at the center of layer bands.

#### 2.2.3. Tensile Strength

The tensile test is one of the most common tests used for material evaluation. The tensile test is performed in its easiest way by gripping the other ends of a test part (specimen) within the testing machine loading frame. In tensile tests, the test specimen will be pulled slowly. After that, the specimen will lengthily stretch, and the tensile may be increased steadily until the test specimens are fracture. The forces-extension data are normally monitored and recorded during the process, which is a quantitative measure of how the test part deforms under the applied tensile force. The stress and strain values are recorded, and the curves are plotted. The tensile testing according to ASTM is E8 [29] for metals, etc. ASTM E8 describes tensile testing of metals and is the actively used standard for the testing of metals. The sample used for testing will have different characteristics for metal, and may be made of sheets or bars. The dimensions of the tensile specimens in mm are shown in Figure 7.

#### 2.2.4. Microstructure

We prepared the workpiece to observe the microstructure and microstructure picture of the hot-wire plasma welding process with a LEXT 3D measuring microscope OLS4100 (Olympus, Waltham, MA, USA). The microstructure analysis samples were polished with the sandpaper grades 120, 400, 600, 800, 1000, 1200, and 2500 grit by machine grinding and polishing. At the micro-level, we used the 50×, 200×, and 500× magnification to perform a complete visual analysis and observation.

### 2.3. Analysis of the Physical and Mechanical Properties

The physical and mechanical properties study in this research were performed in four steps (porosity, Vickers hardness, tensile strength, and microstructure), and the workpiece was divided into parts as shown in Figure 8. The workpiece was cut with a wire cut EDM (electrical discharge machine, model: AQ325L, Sodick, Schaumburg, IL, USA) with a band saw and using a slow cutting speed.

## 3. Results

### 3.1. Parameters and Welding Process

The welding found that sample number 2 (Figure 2) had a better appearance and shape than the other weld bead and met the requirements. The deposition of the hot-wire PAW process from sample number 2 is shown in Figure 9 and Figure 10. These parameters were constant: a welding speed of 1.83 mm/s, an arc current of 120 A, a wire feeding speed of 0.85 m/min, and a wire current of 35 A.

Figure 9 showed the final arc voltage value from 1 layer to 36 layers of the deposit process. The arc voltage gives a clear indication of arc length at a given current. These systems directly measure arc voltage and control the torch height to correct the error. It may be defined as the arc voltages that appear across the contact during the arcing period when the current flow is maintained in the form of an arc. In addition, arc voltage depends on the shielding gas, current, electrode angle, workpiece composition, arc blow, and wire feeding, and it must be set for the particular conditions being used. The arc voltage rises and falls per layer because the arc voltage values are measured from the electrode tip to the melting pool, which leads to the results from the machine. In the experiment, it was found that the arc voltage was not clearly controlled, the last weld sagging (falling) causing the arc voltage to rise and instability. Arc voltage values will be measured during the arcing, and the resulting values change with the arc distance. These arc voltage values are the final arc voltage that occurs after the welding of each layer.

Figure 10 shows the temperature values of the deposition of the hot-wire PAW process per layer (all 36 layers). There were temperature measurements for the weld bead: infrared thermometer (IR) measures the temperature of the thermal radiation of the object, and can typically measure temperature in a range of −25 °C to +380 °C (14 °F to 716 °F). The temperature was measured on the surface of each deposited layer at the mid-point of each welded bead layer. After finishing each deposited layer, the deposition process resumed after five-minute break. The interpass temperature was not constant because the heat generated by the arc voltage was not constant. The fluctuation of the arc voltage during the deposition process was due to the fluctuation of the arc gap distance.

The values of the bead height titanium alloy walls built by the hot-wire PAW process from sample number 2 are shown in Figure 11.

Figure 11 showed the values of the bead height titanium alloy walls by the hot-wire PAW process (36 layers). The dark blue line showed the height of the weld bead in Position 1. The green line showed the height of the weld bead in Position 2. The pink line showed the height of the weld bead in Position 3. The light blue line showed the height of the weld bead in Position 4. The yellow line showed the height of the weld bead in Position 5. Meanwhile, the orange line and red line showed the height of arc distance and an average of bead height, respectively. There were two main measurements for weld bead height: (1) The vernier caliper is a precise tool that can be used to accurately measure outside diameter, inside diameter, and depth. (2) The welding gauge is the main scale, height gauge, and undercut depth gauge. It is a welding inspection, for a variety of bevel angles detecting weldments, height, width, gap, and undercut depth.

Moreover, the arc distance is the layer-by-layer movement of the robot in the build direction. From the first layer (start), the arc distance was the distance between the torch and the baseplate at a height of 0 mm. After that, we moved the robot according to the average height of the welded bead in each layer. Therefore, the arc distance increases linearly.

### 3.2. Porosity Analysis

In the case of titanium alloy, there is always a certain proportion of porosity, and the content significantly depends on the use conditions [34,35]. Figure 12 shows porosity in a sample specimen in the AM part: (a) front area; (b) back area. The porosity formed in the titanium alloy samples produced over a range of melt scan speeds, from 100 to 1000 mm/s, were investigated in [36].

The pore sizes in Figure 12 were measured using computerized tomography scans (CT scans) and a measuring gauge. A higher density of porosity was mainly distributed at the beginning of the welding process. The volume of pores measured in the part was 47 porous in the consolidated material, per area at a scale of 21:1 mm, but the number density of the pores was lower in the back region. The front area had a maximum porous size of 1.129 mm, and a minimum porous size of 0.45 mm. The back area had a maximum porous size of 0.44 mm, and a minimum porous size of 0.25 mm. In the front area, the porosity occurred the most because it is often opened by the shielding device as welding begins. After several trials building walls, the shielding gas cover was improved, the gas content was increased, and the porosity was reduced.

### 3.3. Vickers Microhardness

The tested Vickers microhardness coupons were incised from the formed thin wall, with a steady value of approximately 206.21–212.73 HV from 15 mm (Position 3), 204.61–204.61 HV from 30 mm (Position 2), and 176.15–182.66 HV from 45 mm (Position 1) from the base plate to the top surface (the starting point of deposition) and were along the additive direction. These results are summarized in Table 5.

As reported for titanium alloy grade 2, US specs (AMS 4951), the average Vickers microhardness values of the deposited layers was around 160–200 HV [27,28]. The average of all the results was 180.05–210.00 HV. When compared with all hardness values, we found that the area with the least hardness was at the top area. The table shows that the hardness values are similar and consistent with the standard values of titanium.

For testing the surface hardness, a microhardness indenter was used. The edge effect was also reported, which may result in lower hardness values. This work involved nano-hardness indentations performed very close to the surface [37]. A maximum microhardness value of micro plasma wire deposition material of 616 HV in the heat effect zone was mentioned in a previous study [38]. The hardness results of the three specimens are plotted in Table 5. From the table, for all specimens, hardness decreases with increasing deposit height. Specimens 1, 2, and 3 were completed with a waiting time of 10 s, and all specimens were completed with 200 g force. The average deposit hardness for Specimen 3 was greater than that of Specimen 2 and Specimen 1.

### 3.4. Tensile Strength

The tensile test is one of the most common tests used for material evaluation. In this paper, we used random sampling for the tensile test. The sampling positions of each wall were sectioned. Three tensile test samples along the vertical direction were equidistantly taken from the middle to the end of the wall. Another three tensile test samples in the horizontal direction were evenly taken from the top to the root of each wall [39]. Tensile testing perpendicular to the build direction produces significantly reduced ductility in comparison to testing along the build direction [33,40]. The tensile strength of the deposited titanium alloy in the vertical directions was less than it was in the other directions [29,31]. Tensile tests for specimens were carried out on a Universal Testing Machine: Instron 8872 (Instron, Norwood, MA, USA). The results of the tensile strength tests for the as-deposited hot-wire plasma welding are shown in Figure 13.

Figure 13 are presents ten-row plots of the tensile strength values obtained. It shows engineering stress–strain curves of the specimens after hot-wire plasma welding (all 10 specimens). The tensile deformation curves show extensive plastic deformation. Graph numbers 1 and 2 show the tensile strength values of the substrate range: 495.63 MPa and 496.65 MPa. For all rows, the tensile strength steadily decreased significantly in the heat-affected zone (HAZ), where the values are mainly in the range of 458.43–578.86 MPa from the bottom to the top part of the deposited sample (graph numbers 3 to 7). The transverse zone is characterized by tensile strength values from 426.42 MPa to 429.69 MPa (graph numbers 8 to 10). The tensile strength values of the transverse direction were worse than those of the longitudinal direction due to the HAZ of the adjacent deposited tracks acting on each other. The Young’s modulus values of all specimens are between 61.50 and 69.34 GPa. It can, however, be noted that, for these specimens, graph number 3 has the highest tensile strength value compared to other graphs, since the first layers inherit the original fine-grained structure of the substrate, so their strength is higher than it is in the rest of the sample.

The sample of the fractured specimen is shown in Figure 14, which indicates that the rupture occurred, and that the specimen suffered acute forming before fracturing. Ten pieces were examined, and they all behaved similarly. In addition, the average tensile strength value of titanium alloy grade 2 is 485 MPa [29]. When compared with the standard tensile strength, the tensile values were similar and consistent with the standard values of titanium alloy grade 2.

### 3.5. Microstructure

The analysis of the deposited microstructure showed a number of features. The staggered individual deposited tracks could be observed at low magnification. Figure 15 and Figure 16 present the as-deposited microstructure in a cross-section vertical to the plasma scanning direction. The weld metal deposit grains change from having a relatively fine microstructure and a small grain size near the substrate to a very coarse microstructure with a large grain size as the distance from the substrate increases.

We used 3D measuring laser microscope OLS4100 (Olympus) analysis to evaluate the microstructure of hot-wire plasma deposited titanium alloy grade 2. To study the phase transformations of titanium alloy grade 2 during hot-wire plasma metal deposition, we transversely cut the sample, polished them with silica suspension, and examined them with an optical and 3D measuring laser microscope.

Titanium alloy has a two-phase (α + β) microstructure. Titanium undergoes α to β phase transformation. In Position 1, the microstructure becomes relatively fine with a small grain size. In Figure 16 (Positions 1–6), the darker regions are the β phase, which remains between the α plates that have developed. The microstructure consists of parallel plates of α characterized by the β phase between them. The microstructure of the substrate was characterized by alpha (lighter part) and beta grains (darker parts). Figure 16 (Positions 4–6) also shows the microstructure in different areas of the top surface of the sample. As can be seen the microstructure varies in different areas. Some areas (Positions 4 and 5) contain mainly primary α grains surrounded by a coarse β phase. The primary α forms through nucleation and growth during the α + β working operation, and its morphology can vary from elongated plates in lightly worked material to equiaxed globular morphology in heavily worked material. The microstructure once again becomes very coarse with a relatively large grain size near the top region of the weld metal deposit and becomes relatively uneven, and expansion of the dendrite is found.

## 4. Discussion

### 4.1. Porosity Analysis

The pore or cave pocket is a characteristic feature of welding and occurs frequently during the welding process. Gas comes out of solution in the form of bubbles since the solubility for gas decreases in a liquid metal upon cooling [41]. Porosity will be increased because the dendritic solidification interface and some inclusions can be used as heterogeneous particles for pores. Pores serve as stress risers that are brittle fractures and that increase the susceptibility to failure [39]. Several studies have shown that advanced tools can help detect faults or other defects that occur in the welding process such as ultrasonic non-destructive testing (NDT) techniques [30], X-ray CT methods [35,42], and radiographic testing (RT) [43,44]. The slow welding speeds and the welding in the flat position, or uphill in the vertical position, encourage the escape of pores [45]. The variation in the arc length and the arc distance was large enough to affect the shielding gas (Argon), and porosity started to show in some layers.

### 4.2. Vickers Microhardness

The decrease in hardness with increasing deposit height is a direct result of the observed increase in dendritic structures with increasing deposit height. The mechanisms differ between the α-phase and β-phase [46]. During the deposition, the melted area can be hardened by a solid solution (contamination from the atmosphere or base material), dislocation, or boundary hardening (a smaller grain or α-colony size, phase transformation, or a high dislocation density [47]). A hardness gradient at deposits is generally expected, as each layer has a different thermal history. The hardness difference between the top region and bottom region is not equal in height. The hardness of the top layers (Position 1) is below more than other layers. In addition, the total heat induced is larger and homogeneously distributed through the specimen. This leads to a heterogeneous material and finally to a slight decrease in hardness from top to bottom. Therefore, the only three measurements (very few) are strongly dependent on the amount of α- and β-phase at the particular location.

### 4.3. Tensile Strength

The location used to measure the longitudinal tensile strength, the transverse tensile strength, and the baseplate of hot-wire PAW for AM deposited titanium alloy grade 2 are presented in Figure 6. The transverse direction refers to samples taken across the layers of the building, while the longitudinal direction refers to those taken along the layers [48]. Figure 13 shows the tensile properties of the specimens deposited by the hot-wire method, which combines plasma welding. Specimens for the tensile test were machined from the bottom to the top of the deposited sample. The test results showed a tensile strength of approximately 426.42–578.86 MPa. The tensile strength steadily declined from the bottom to the top part of the deposited sample. The results reveal that the percentage elongations of all specimens remained almost constant (14.46%). Compared with the longitudinal direction, the structural solidarity of parts in the transverse direction was worse due to the HAZ of the adjacent deposited tracks acting on each other. The voids (porosity or crack) were the main cause of the bad tensile strength in tensile specimens. Moreover, the nearby, inadequately fused areas were torn away, and the cracks then extended. At the same time, voids reduced the effective sectional areas of tensile specimens, and the tensile strength was worse [45]. The coating was able to increase the tensile strength of the test part surface, which was mentioned in [49]. Therefore, the tensile strength and elongation in the longitudinal direction were higher than those in the transverse direction.

### 4.4. Microstructure

The analysis of the deposited microstructure showed a number of features. The staggered individual deposited tracks could be observed at low magnification. The formation of the microstructure and the phase of titanium alloy occur as a result of melting at 1900 °C and subsequent rapid cooling [50]. Figure 16 presents the as-deposited microstructure in the transverse section of the baseline at deposit heights. Regarding the microstructure of the specimen near the surface, the light-colored regions of the microstructure are the alpha grain, and the dark regions are the beta grain. It was observed that the prior β-grains and α-grains have a globular shape at the bottom, i.e., near the base and the substrate. In addition, with increasing height, the microstructure of the specimen near the top region had many large dendritic structures, and the number of grains decreased. The heat input led to the growth of these grains in the build. The microstructure of the titanium alloy deposited alloy depends on the time at the temperature, the peak temperature, and the cooling rate of each layer of deposited material during the multiple thermal cycles. As regards the microstructure of titanium-based deposits [51], the needle-like dendritic microstructure of alpha-phase titanium was found to develop from bead formation, along with rapid cooling of the melt. According to [46], the main microstructural features of the welding process are, in addition to the β and α grain size, the conditions under which the high-temperature β and α phases transform during cooling. During the plasma welding, the heat flow direction of the solidification of the molten pool was about perpendicular to the surface of the substrate or the pre-deposited layers. The plasma transfer arc-assisted deposition technology can be also used to directly manufacture a real component [52]. In [53], it was found that the microstructure of the sample was mostly a fine columnar dendritic structure and that grew epitaxially along the deposition direction. The growth rate, temperature gradient, melt pool shape, welding speed, and the alloy constitution will all control the final microstructure of a solidifying melt pool in AM [50].

## 5. Conclusions

The following conclusions can be drawn based on the analytical and experimental investigations.The variations in welding speed and wire feeding speed that resulted from the in-consistent angular velocity led to inconsistent overall deposit geometry. To develop reliable hot-wire PAW process parameters, an accurate welding speed, and wire feeding speed must be determined for consistent weld metal deposit build-up. Nevertheless, a moving speed of the welding that is too great, could cause humping. The appropriate hot-wire plasma welding parameters are a welding speed of 1.83 mm/s and an arc current of 120 A. A hot-wire current (amperage) of 35 A, a wire feeding speed of 0.85 m/min, and a reverse feeding speed of 5 m/min (0.99 s) gave the appropriate hot-wire temperature in the weld pool range. In a single run, wall widths were approximately 7–13.25 mm after welding.The Vickers microhardness of as-deposited samples was in the range of 180.05–210.00 HV and the tensile strength was 426.42–578.86 MPa depending on the orientation and location of the specimens. The microstructure evolution, tensile strength, and yield strength decreased gradually from the bottom to the top part of the deposited sample because of the observed increase in grain size with increasing deposit height. The percentage elongations of all specimens were at an average of 14.67%. This indicates that the as-deposited sample exhibits improved mechanical properties for the hot-wire PAW welding process.The microstructure of the specimen near the bottom region (or substrate) consists of alpha (lighter part) and beta grains (darker parts). Cross-sections of the weld metal specimens showed an overall increase and a large size in the dendritic structure (top region of the part). The dendrite spread out until all solid metals and then stopped growing. This caused the alloy to enter into a non-equilibrium condition because the hot metal alloy cooled instantly or too quickly and suddenly solidified, whereas the inside remained hot and soft. Furthermore, the coarsening microstructure, as the height of the weld metal specimen increased, grew epitaxially along the deposition direction. However, there was not little effect found on the strength. Each crystal was often unequal because the growth of each dendrite was independent.The titanium alloy is highly susceptible to oxidation during welding. Oxidation and distortion can cause problems in particular when the deposition takes place outside of the chamber. During the manufacture of the first weld, a metal specimen found oxidation was observed.Further experiments are necessary to optimize technical parameters for a suitable deposition condition and improve the quality of the hot-wire PAW process.


## Figures and Tables

**Figure 1 materials-14-01270-f001:**
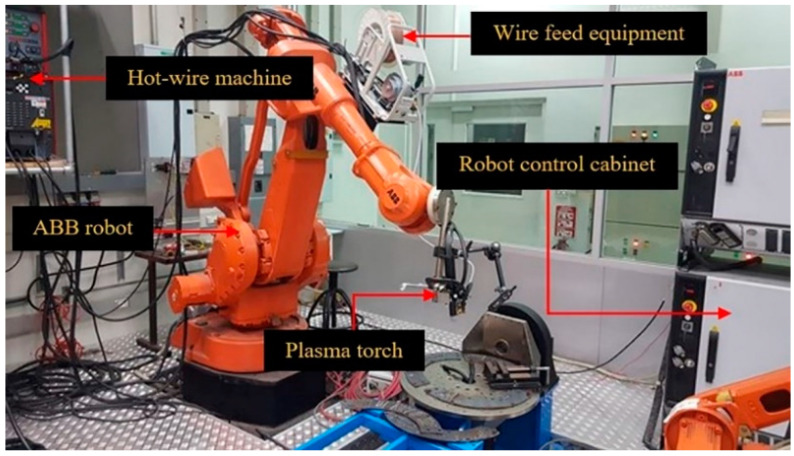
Setup of the hot-wire plasma welding for the additive manufacturing (AM) process system.

**Figure 2 materials-14-01270-f002:**
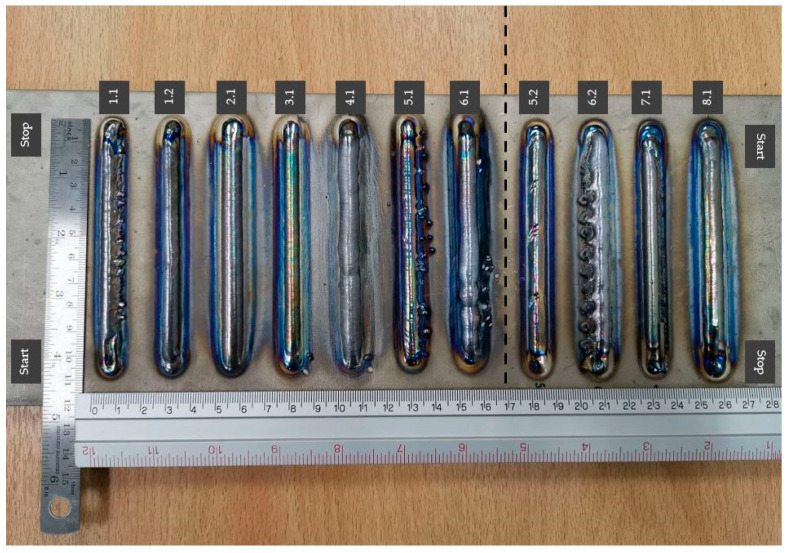
Deposited single beads for a designed experiment for the hot-wire plasma welding process (Numbers 1–8).

**Figure 3 materials-14-01270-f003:**
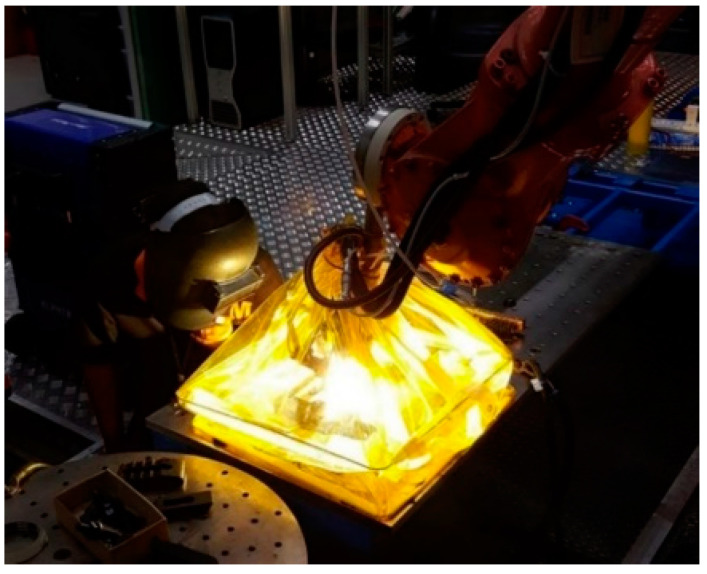
The flow-purge welding chamber.

**Figure 4 materials-14-01270-f004:**
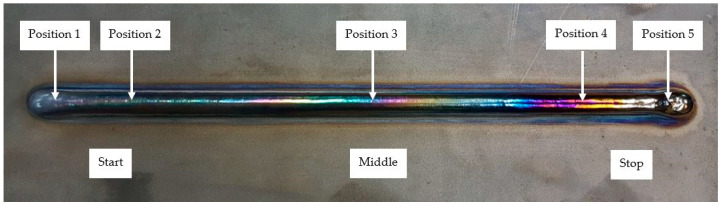
Positions of the weld bead height and bead width measurement for the hot-wire plasma welding process.

**Figure 5 materials-14-01270-f005:**
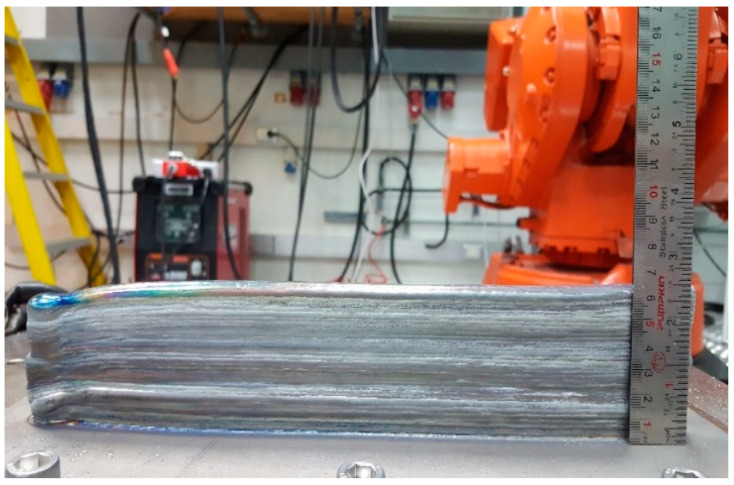
Photographs of the deposited walls: 36 layers.

**Figure 6 materials-14-01270-f006:**
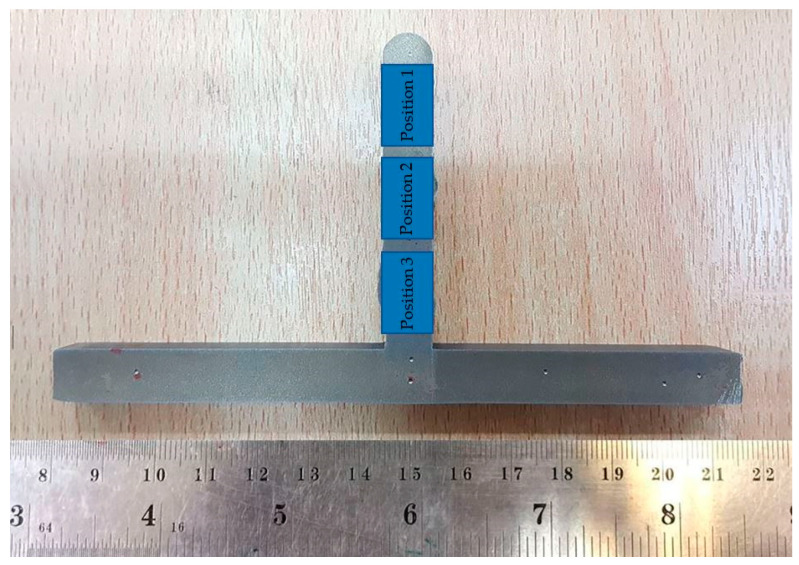
Schematic illustration of microhardness measurement on the profile of the microhardness of layer bands.

**Figure 7 materials-14-01270-f007:**
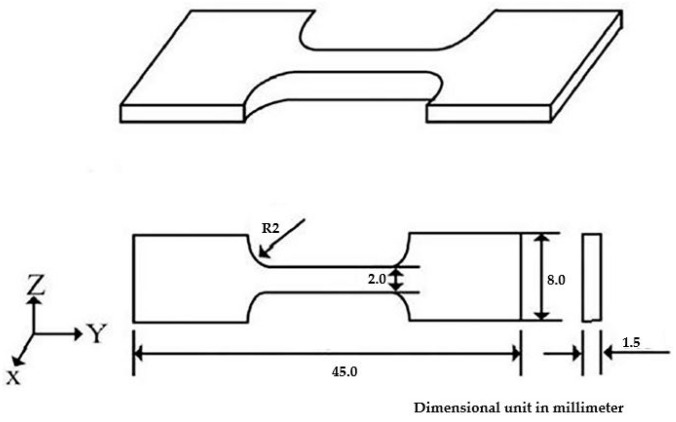
Dimensions of the tensile specimens (mm).

**Figure 8 materials-14-01270-f008:**
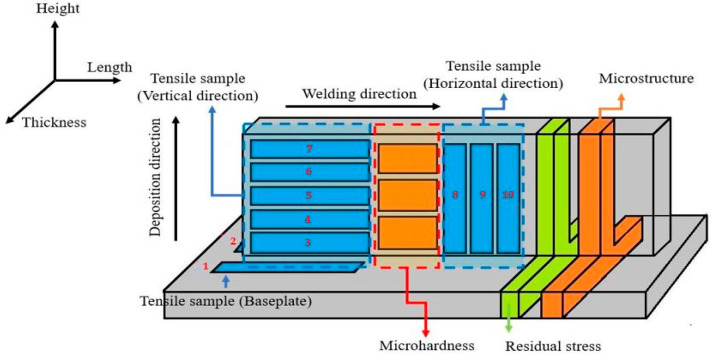
Each part of the workpiece.

**Figure 9 materials-14-01270-f009:**
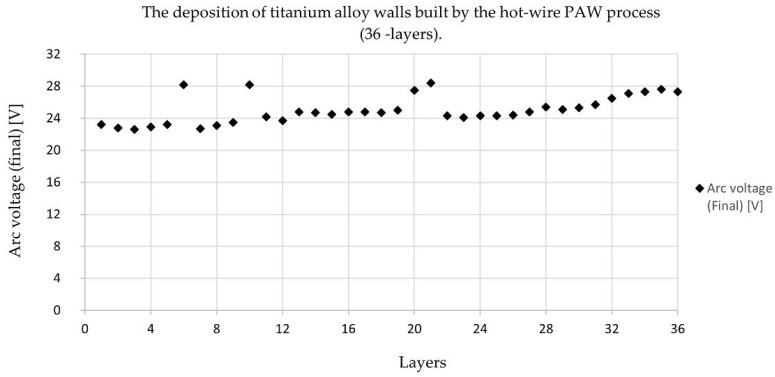
Arc voltage (final) values of the deposition of the hot-wire PAW process per layer.

**Figure 10 materials-14-01270-f010:**
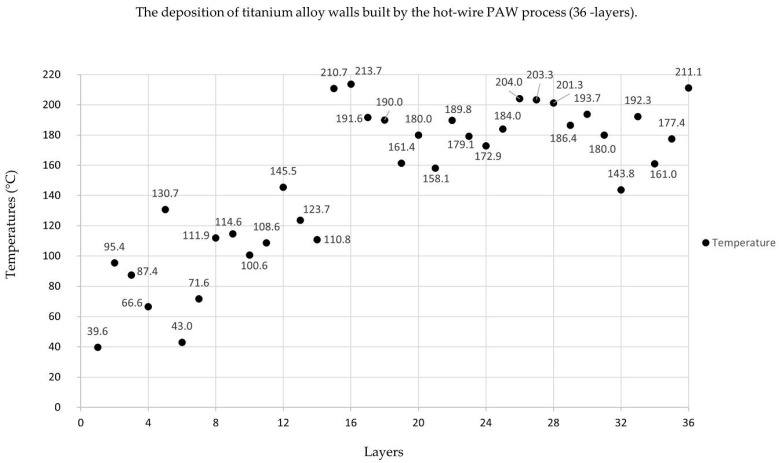
Temperature values of the deposition of the hot-wire PAW process per layer.

**Figure 11 materials-14-01270-f011:**
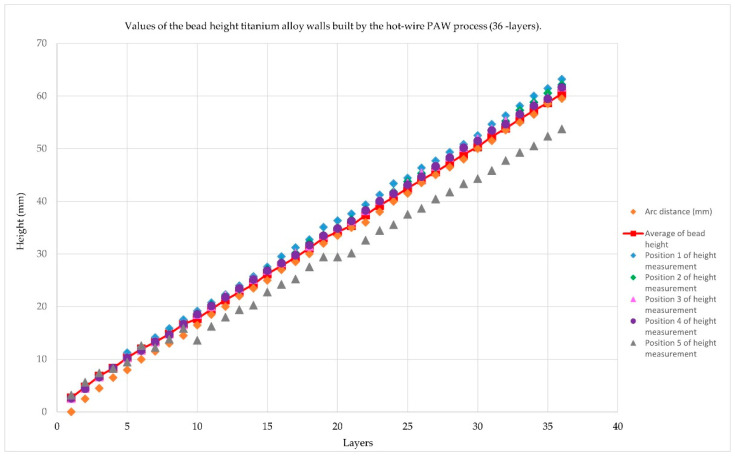
Values of the bead height titanium alloy walls built by the hot-wire PAW process (36-layers).

**Figure 12 materials-14-01270-f012:**
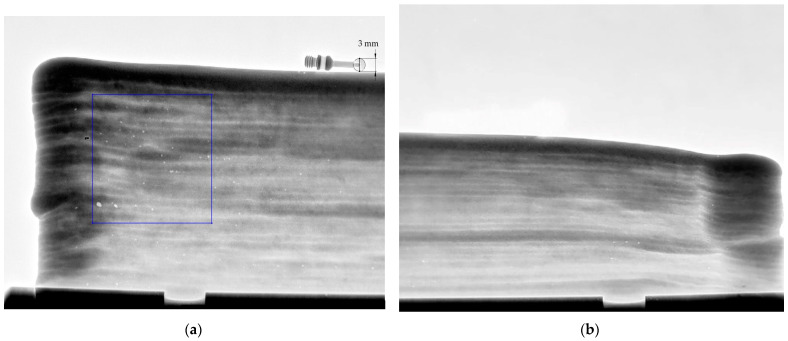
Defect volume of distribution of hot-wire plasma welding: (**a**) front area; (**b**) back area; scale 21:1 mm.

**Figure 13 materials-14-01270-f013:**
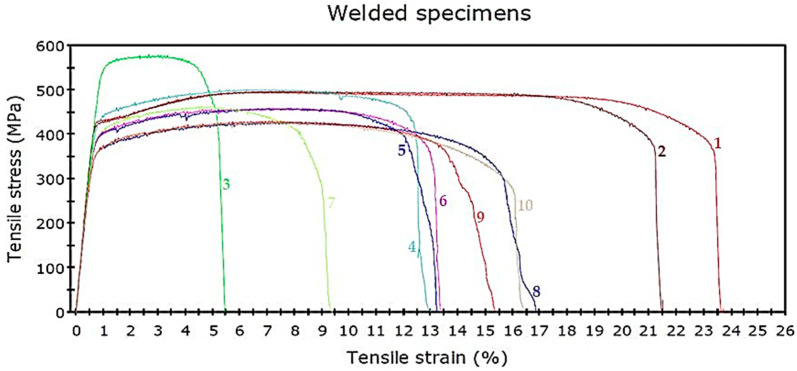
Engineering stress–strain curves of the specimens after hot-wire plasma welding.

**Figure 14 materials-14-01270-f014:**
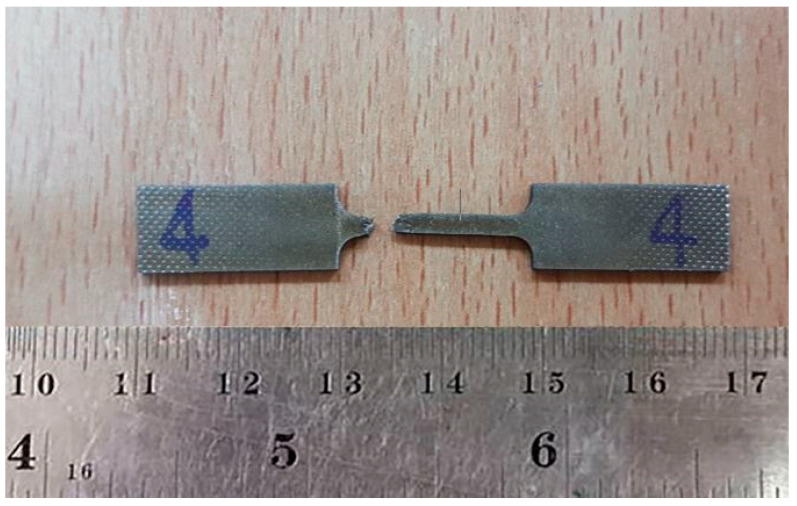
The tensile test specimens (Position 4).

**Figure 15 materials-14-01270-f015:**
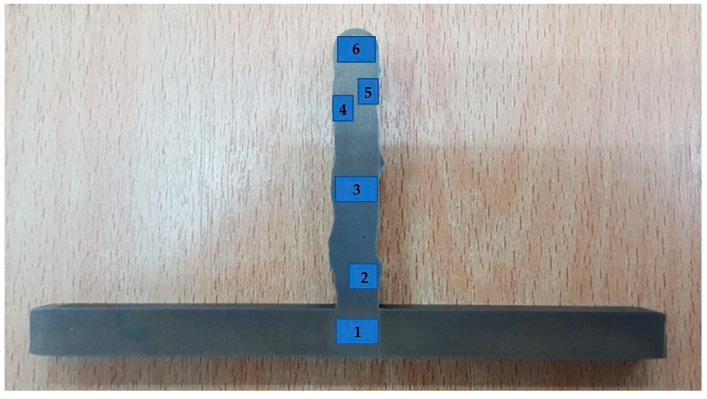
Position of the microstructure measurement of the transverse section of the specimen.

**Figure 16 materials-14-01270-f016:**
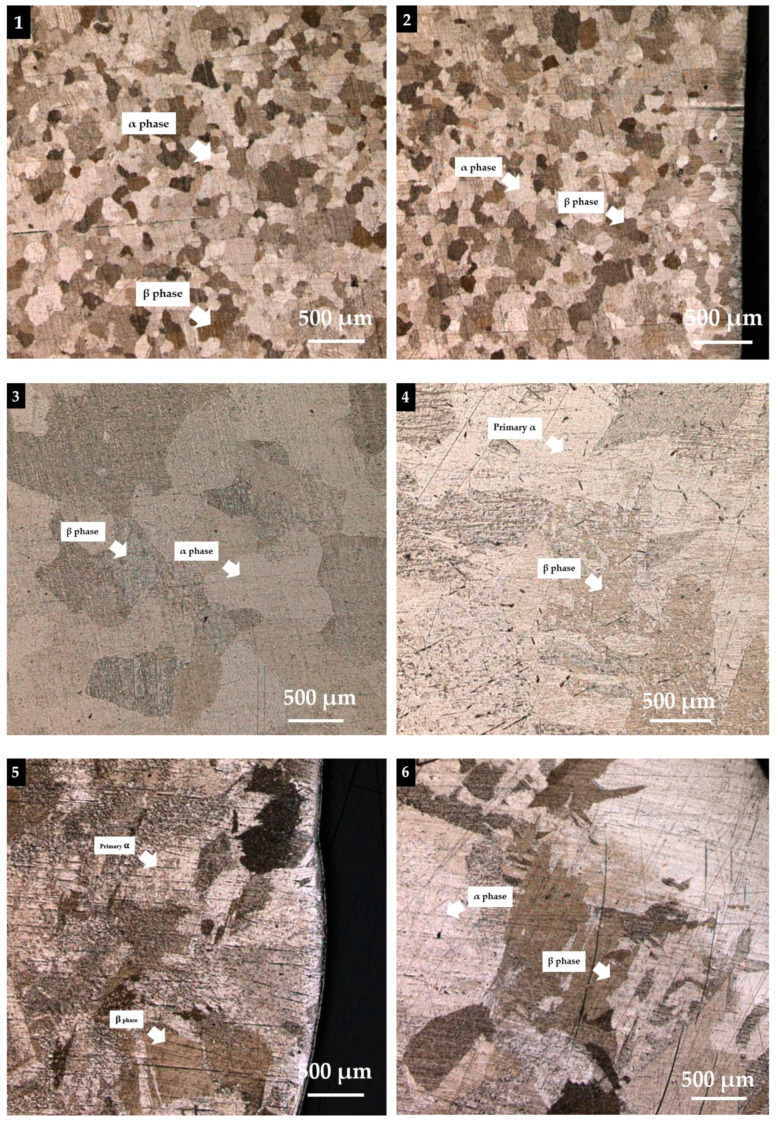
Microstructure of (**1**) Position 1, (**2**) Position 2, (**3**) Position 3, (**4**) Position 4, (**5**) Position 5, and (**6**) Position 6 of the hot-wire plasma welding test.

**Table 1 materials-14-01270-t001:** The chemical composition of the titanium alloy grade 2.

Elements	Standard (% w/w)
N	0.03
C	0.08–0.10
H	0.015
Fe	0.30
O	0.25
Iron	0.30
Ti	Balance

**Table 2 materials-14-01270-t002:** The mechanical properties of the titanium alloy grade 2.

Mechanical Properties		
Mechanical Properties	Standard Value	Units
Average tensile strength	485	MPa
Yield strength (0.2% offset)	275–448	MPa
Elongation (% EL)	18–20	%
Hardness Vickers	160–200	HV
Modulus of elasticity	103	GPa

**Table 3 materials-14-01270-t003:** The welding experimental conditions of eight welding specimens for hot-wire plasma welding.

No.	Welding Speed [mm/s]	Arc Current [A]	Arc Voltage: Final [V]	Heat Input [kJ/mm]	Hot-Wire		Reverse Feed Speed	
					Wire Feed Speed [m/min]	Wire Current [A]	m/min	s
1.1	3.00	120	21.6	0.518	1.70	35	5.00	0.99
1.2	3.00	120	22.2	0.533	1.70	35		
2.0	1.83	120	23.4	0.921	0.85	35		
3.0	3.00	120	23.9	0.574	0.85	30		
4.0	1.83	120	24.7	0.972	1.70	35		
5.1	3.00	120	24.0	0.576	0.85	35		
5.2	3.00	120	21.3	0.511	0.85	35		
6.1	1.83	120	25.0	0.984	1.70	30		
6.2	1.83	120	20.5	0.807	1.70	30		
7.0	3.00	120	20.6	0.494	1.70	30		
8.0	1.83	120	20.6	0.810	0.85	30		

**Table 4 materials-14-01270-t004:** The parameters used for the hot-wire plasma welding process.

Parameters	Details	Unit
Welding speed	0.85–3.00	mm/s
Wire feeding angle	38	deg
Wire feeding position	1	mm
Stand-off distance/Arc distance	6	mm
The flow rate of shielding gas	20 (Ar)	l/min
Arc current	120	A
Wire current (Amperage)	30–35	A
Wire feeding speed	0.85–1.70	m/min
Pilot arc	20	A
Gas plasma	0.5	l/min
Gas shield	20	l/min
Plasma gas flow rate	0.5	l/min

**Table 5 materials-14-01270-t005:** Results of the Vickers microhardness test for hot-wire plasma welding.

Sample	Positions			Average	Standard Deviation (SD)
	1	2	3		
Position 1	182.66	181.33	176.15	180.05	2.80
Position 2	204.61	206.11	204.61	205.11	2.70
Position 3	212.73	211.06	206.21	210.00	1.70

## Data Availability

The data presented in this study are available on request from the corresponding author.

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
