# Peer review of "Experimental Investigation of Additive Manufacturing Using a Hot-Wire Plasma Welding Process on Titanium Parts"

_materials, 2021, doi:10.3390/ma14051270_

Round 1

Reviewer 1 Report

As part of the research, walls were additively manufactured from a Grade 2 titanium. A hot wire plasma process was used for this purpose. The contents of the paper need a thorough revision to allow an understanding of the implementations and the resulting outcomes. The introduction is generalized and specific information about the material, its processing (affinity to atmospheric gas absorption, gas shielding, etc.) and application areas are missing. The presentation of results is not fully comprehensible and needs improvement. The summary is generalized, no specific connections are shown.

Point 1: page 2 / Line 57-60 Why is it important for the additional wire to be melted independently of the arc?

Point 2: page 2 / Line 57-60 Confusion / partial repetition of facts. Out of context.

Point 3: page 2 / Line 68 What permeability is meant here?

Point 4: page 2,3 / 51-104 The text content is difficult to follow and littered with repetitions. Please create a stringency.

Point 5: chapter 2 Please specify the dimensions for the welded wall structures.

Point 6: page 3 / 118 What is a “6-axis ABB robot of freedom”?

Point 7: page 3 / 120-121 Why this Information from source [35]?

Point 8: chapter 2.1 Provide more details about the experimental setup. Why is the hot-wire machine “controlling the hot-wire plasma welding path” (page 4/131)? What is meant here?

Point 9: page 5 / 145-147 What are the applicable criteria other than a better appearance and shape?

Point 10: page 5 / Figure 2 Very strong tarnish colors. How was the residual oxygen content? Which protective gas flow was set?

Point 11: page 5 / Figure 3 What is the purpose of the illustration? What can be seen here? So far unclear.

Point 12: page 6 / 165-168 Recurrence of the content.

Point 13: page 7 / Figure 6 Very strong tarnish colors. How was the residual oxygen content?

Point 14: page 7 / 180 There should be another term for the “surface integrity study”.

Point 15: page 7 / 182-184 Not a meaningful sentence, please rephrase.

Point 16: page 8-10 / Table 5 and 6 The relevant content of the tables can be presented in a more scientific and representative way, e.g. in the form of diagrams.

Point 17: page 8 / Figure 8  The porosity can be seen conditionally in figure 8. Figure too small.

Point 18: page 9 / 208-214 What is the total porosity (e.g. area, volume)? To what extent was the porosity reduced?

Point 19: page 9 / Table 5 Why does the voltage rise and fall per layer? Is this due to a non-constant electrode distance? How was the adjustment made in the buildup direction per layer? Was welding carried out without a waiting period or was there a time-controlled welding interlayer pause? There is no constant interpass temperature.

Point 20: page 10 / Table 6 Why does the arc distance increase linearly? Is this the layer-by-layer movement of the robot in the build direction?

Point 21: page 11 / Table 7 and 8 Why are the raw data given? Why are all results presented in tabular form so far?

Point 22: page 11 / 240-247 Confusing description. Please revise and add schematic illustration for better understanding.

Point 23: page 12 / Figure 9 In the previous understanding, the diagram does not make sense. Why does the average increase, although the positions have an almost constant course. What does the average value belong to?

Point 24: page 12 / 253 I couldn’t find the hardness values in source [34].

Point 25: page 13 / 268 Why is the tensile strength of Ti-6Al-4V listed here?

Point 26: page 13 / Table 9 Again, why table form and no diagram?

Point 27: page 13 / Figure 11 The diagram has no physical relevance. All curves should start at an elongation of 0%. Major scientific mistake.

Point 28: page 14 / 285-286 How can you recognize in picture 13? Please add an enlarged picture and a scale.

Point 29: page 15 – 17 / discussion Many sources are cited but there is no substantive discussion of the results. In some cases, errors due to powder residues are mentioned, which have no relevance to the paper. Mostly a conclusion is drawn to Ti-6Al-4V, although titanium grade 2 is used in the investigations. Complete revision of the discussion is recommended.

Point 30: page 17 / 409-410 The sentence makes no sense.

Point 31: page 17 – 18 / conclusion What are the concrete results? What is the added value of the studies? For example, it is widely known that titanium has an affinity for atmospheric gases.

Author Response

We would like to thank you for the reviewer for a careful thorough reading of this manuscript and for constructive comments. We have considered all the comments highlighted by the reviewer and revised our manuscript accordingly. Please find our responses to your comments on pdf file atteach.

Reviewer 2 Report

Dear Authors,

I have reviewed paper titled "Experimental Investigation of Additive Manufacturing using a Hot-Wire Plasma Welding Process on Titanium Parts". The topic is interesting and very important, so it could be considered for potential publishing.

Your manuscript fulfills the aims and scope of Materials. My suggestions are listed below.

General remarks:

  • Please check your references' list.The style is different to the journal template (https://www.mdpi.com/journal/materials/instructions).

Introductions:

This part presents relevant scientific background. I have some minor comments here:

  • I propose change the title of this paragraph to "Introduction", because you have presented one part for one manuscript.
  • Lines 72 and 78 - these references are important (as you stated by style of citations), so I propos to change the description to "Name et al. [X]...".

Experimental procedure and materials:

  • In this section you have described materials firstly. I propose change title to "Materials and Experimental Procedure".
  • Table 3 - please add values of heat input [kJ/mm]. Also, you can deletelast 0 in Arc Voltage column. The "Reverse feed speed" parameters are constant, so I propose to mark this in th text and delete relevant columns.
  • Fig. 2 - please add scale bar.
  • Lines 165-167 - duplication of lines 148-150.
  • Have you used any requirements from standard during hardness measurements? If yes, please mark the relevant standard here.

Results:

  • Please move the description of methodology used during X-rey test to the previous paragraph.
  • Fig. 8 - please change the size of numbers. Now, it is hard to see the values.
  • Lines 221-231 - this is description of methodology, and should be moved to previous paragraph.
  • Please add information about schematic hardness measurment points distribution. Now it is hard to see ,in which places you have measured the hardness. Also, the information about standard deviation should be marked for avarage values.
  • The description of tensile strength methodology is missing. Have you used static tensile test? Any requirements from standards were used? Please add this information to the methodology (previous paragraph).
  • Lines 298-303 - this is description of methodology and should be moved to previous paragraph.

Discussion:

Conclusions:

  • These sections are the strongest in your paper. You have compared your results with the literature very well. Also, the conclusions are strongly connected with results. Congratulations.

Following the high-level discussion and quite well description of results.

Author Response

(The authors gave the same response as above.)

Reviewer 3 Report

Journal: Materials

Manuscript ID: materials-1096861

Title: Experimental Investigation of Additive Manufacturing using a Hot-Wire

Plasma Welding Process on Titanium Parts

Authors: Pattarawadee Poolperm, Wasawat Nakkiew *, Nirut Naksuk

The paper investigates a modification of additive manufacturing. The plasma deposition process is improved by hot wire. These technologies definitely have relevance. It is the wire technology that is especially interesting, because it makes it possible to achieve a high printing speed. Commercially pure titanium was chosen as the material. The choice of this material is clear. It is a simple single-phase material, quite popular in the industry and well studied. For the beginning of technology development, this alloy is well suited, since there are really few works with this combination of technologies. For this reason, the subject matter of the work definitely has novelty and may be of interest to the journal audience. However, the methodology raises questions. In fact, the work selects welding parameters. The quality of the samples is evaluated visually. And then all the studies are carried out on one sample. This approach does not give an opportunity to assess the advantages of the technology.And the quality of the selected mode does not give a set.  The quality of the work done is also no good. Studies are conducted, data are obtained, but not analyzed properly, no real scientific knowledge is obtained. Discussions of the results often contain general arguments. Illustrative material is redundant, which visually increases the size of the paper. The remarks are presented in more detail below:

  1. Table 2 is redundant. First, Tensile strength and Ultimate strength are the same thing. Second, it is generally accepted that the metric system is used. Imperial units are obviously excessive here. Third, the article does not perform Charpy tests, Fatigue tests and HRB measurements and is not even discussed in any way. That is, these data are redundant within the framework of the article. Fourth, you should specify the range, not the minimum value. The average strength of this alloy is 485 MPa. By stating the minimum value, you can mislead the reader.
  2. The substrate material is not listed anywhere.
  3. Unfortunately, it is impossible to distinguish anything in Figure 3.
  4. Lines 164-167 copy lines 147-150.
  5. You got a nice sample of the wall, but there is no need to duplicate it in Figure 6. One picture is enough.
  6. Tables 5 and 6 are not discussed in the text at all. What do they mean and why are they given?
  7. Table 5. What is this temperature? How was it measured?
  8. Why does the arc voltage increase during the process? It is often the other way around, the first layers require more voltage to heat the substrate.
  9. How was the height of the bead measured? What is Arc distance and why does it increase?
  10. The raw data in the article is obviously unnecessary (Table 7). If you still want to give it, it should be in an appendix, so as not to distract the reader from the meaning. Moreover, it is worth explaining what d1, dm, and so on are.
  11. Figure 9 is clearly redundant. This data is already given in Table 8. You will have to choose what to keep. If you decide to leave the figure instead of the table, you will have to make a correct title.
  12. Figure 10 makes no sense. This is the third time you have given the same data. The literature values of microhardness should be given earlier in the table for comparison.
  13. In the paper you investigate commercially pure titanium. For what purpose on lines 267-269 you mention the alloy Ti-6Al-4V and references 41-42? What sense does this make in the context of the paper?
  14. On lines 272-275 you are again comparing the strength of the products with the strength of TiAl intermetallic alloys, which are irrelevant to the paper. You should compare the values obtained with the strength of your own alloy, which you cited in Table 2.
  15. In Table 9, you should indicate in which direction the samples were placed. For example, enter additional horizontal lines with a notation. It is not necessary to specify two values for the strength of the substrate, because they do not depend on the location, so it is better to specify the average value right away.
  16. Figure 11 is wrong. First, you should make a legend, or sign each line by sample number. Second, how do you have the elongation start immediately at 10%? The graph should go from the point of origin. Third, so many curves doesn't make sense. It is worthwhile to give 3-4 (maximum 5) typical graphs that show the behavior of the material. In addition, the behavior of the material is almost never discussed. The abnormal strength of the sample in position 1 is not explained.
  17. Lines 271-272: "To increase the size and improve the ductility, heat treatments have been attempted." Size of what? Where are these attempts? What did these attempts accomplish?
  18. Figure 14 caption: it appears that you don't mean 50 μm... but rather 50X magnification.
  19. Lines 287-288: you should also give average rather than minimum alloy strength values.
  20. Line 321: you talk about different types of pores. However, these words are not supported by your results. In fact, the pores are only shown in the X-rays, from which you cannot determine the type of pores. How did you determine the type of pores?
  21. Line 344: you are again comparing technically pure titanium to the Ti-6Al-4V alloy. This is incorrect. Grade2 is a single phase material, there is no over aging or aging at all.
  22. Most of the references in the text make no sense. For example, line 341: "The microhardness test indicates that materials (titanium alloy) were softer near the machined surface [50]." What exactly does reference [50] show? Why do you cite it? Every reference needs to be reflexed. For example, as in line 369.
  23. The results of microstructure studies are not discussed at all in the paper. Paragraph 4.4 contains references to the literature with a discussion of general and known issues. Nothing is said about the actual results obtained in this paper. In addition, there is again a discussion of Ti6Al4V, even though it is a completely different alloy.
  24. Line 359: you talk about the relationship between strength and grain size. This is a well-known fact. But you don't have grains, you have a dendritic structure.
  25. Conclusion 1: you are essentially saying general things here that are known without doing any research. In addition, you talk about reducing heat input, and you also compare different technologies in terms of cost. However, in the paper itself, you did not do any such research. You didn't present your results to support the reduction of heat input. The paper only has a study with hot wire. Also, there is no data from you comparing cost to other technologies. Thus, the conclusion is not supported by your results.
  26. Conclusion 3: Here you talk about oxidation for the first time in the entire paper. Oxidation of titanium alloys is a known fact. It has not been investigated or discussed in any way in your paper. This conclusion is not supported by the results of the paper.
  27. Conclusion 5: You claim the presence of columnar grains, although the images do not show them. It looks more like a dendritic structure. In addition, you talk about grain growth. However, there are no measurements of grain size in the entire paper. There are no comparative images from different areas to confirm this. The conclusion is not supported by the results.

Author Response

(The authors gave the same response as above.)

Round 2

Reviewer 1 Report

Almost all comments were addressed by the authors. Some aspects have not been sufficiently clarified from a scientific point of view. Accordingly, further revision and extensive english editing is explicitly recommended before publication. Furthermore, comment 19 (from the first review) contains two fundamental questions. Only one was addressed. The described microstructural evolution is not correct for the used material and especially cruicible for the present journal.

Point 1: page 3 / line 104-105, What is the definition of good mechanical properties? Please clearify. Titan Grade 2 was used in the paper.

Point 2: page 3 / line 121, The authors should use another word for “axis table”.

Point 3: page 3 / line 123-125, In summary, the movement in additive build direction is performed by the robot? Please describe more simply.

Point 4: page 4 / line 135, It exists no brand “Power assist”, this is the model. The manufacturer/brand is probably “MAC”. Please check.

Point 5: page 5 / figure 3, You can't see the process. You can only see different shades of gray. The illustration has no (scientific) value.

Point 6: page 9 / figure 9, Was the additive process carried out without a waiting period or was there a time-controlled welding interlayer pause? Because there is no constant interpass temperature. Figure 9 was insufficiently mentioned and the content was not explained in the text.

Point 7: page 9 / figure 10, I doubt that the heights of the weld beads are plotted in Figure 10. Is this the structure height, since it increases almost linearly with each layer? What is the arc distance?

Point 8: page 12 / figure 14, The legend for the differently colored graphs is missing.

Point 9: page 13 / line 342, What is a “relative microstructure”? Please describe.

Point 10: page 14 / figure 17, Please define more precisely from which area the detailed images were taken. Possibly with a panoramic image of the entire specimen.

Point 11: page 15 / line 403-404, What are the microstructural components of ferrite and pearlite in titanium grade 2? Ferrite and pearlite are phases and microstructural components of iron and steel, respectively. They have nothing to do with titanium. Please provide a source or change it.

Point 12: page 16 / line 440, Again, source for the presence of ferrite and pearlite in titanium or delete/change text areas.

Author Response

We would like to thank you for the reviewer for a careful thorough reading of this manuscript and for constructive comments. We have considered all the comments highlighted by the reviewer and revised our manuscript accordingly. Please find our responses to your comments on the pdf file attached.

Reviewer 3 Report

There are still some comments on the work:

  1. Imperial units remain in Table 2.
  2. Section 3.1: You gave an explanation of why arc voltage can theoretically change. These are general words. However, an explanation is needed as to why exactly it is continuously increasing in your work.
  3. Figure 9: You have not given an explanation of the temperature in the article. In addition, it is still unclear from your answer what this temperature is. Is it the surface temperature of the sample after the layer has been deposited? If so, at what point of it and at what time? It is doubtful that the temperature of the deposited titanium layer can drop to 100 °C in 3 minutes.
  4. There is no explanation of Figure 10. In your response to the comments, you explained how you measured the height. This should be given in the article. Also the article should explain why you did it and what the measurements show.
  5. Image 13 has no meaning.
  6. The explanations in section 3.3 describe the methodology of the measurements. But you have a special section 2.2.2 for that. It is the results that should be described in the "results" section. The same applies to other sections.
  7. I understand that in lines 309-315 you described the results of other researchers. How do they relate to your work? What do they illustrate at all, and why are they given?
  8. There is still no explanation in Figure 14. There is no labeling or captioning of the lines.
  9. Figure 15 should be given in section 2.2.3.
  10. Lines 326-328: If you think the result is caused by an error in the experiments, it should not be given. But I would explain this abnormal strength by the influence of the substrate. It is obvious from the results that the first layers inherit the original fine-grained structure of the substrate, so their strength is higher than in the rest of the sample.
  11. Section 4.2 again describes the methodology and results. Only the last sentence on lines 347-376 is directly relevant to the discussion. The methodology should be described in the methodology section, and the results should be given in the results section.

Author Response

(The authors gave the same response as above.)

Round 3

Reviewer 3 Report

The actual comments have been eliminated.